

# Time-reparametrization invariances, multithermalization and the Parisi scheme

Jorge Kurchan

Laboratoire de Physique de l'École normale supérieure, ENS, Université PSL, CNRS, Sorbonne Université, Université de Paris, F-75005 Paris, France

## Abstract

The Parisi scheme for equilibrium and the corresponding slow dynamics with multithermalization - same temperature common to all observables, different temperatures only possible at widely separated timescales – imply one another. Consistency requires that two systems brought into infinitesimal coupling be able to rearrange their timescales in order that all their temperatures match: this time reorganisation is only possible because the systems have a set of time-reparametrization invariances, that are thus seen to be an essential component of the scenario.

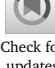

# 1 Introduction

A finite dimensional system whose equilibrium solution follows the Parisi scheme [1] will take an infinite time to reach this equilibrium starting form random configuration. It may also be driven into an out of equilibrium steady-state by an infinitesimal drive, such as shear [2,3], or time-dependence of disorder [4]. If the relaxation times are long, or, in a steady-state, if the drive is weak, the dynamics are slow: this is the regime we are interested in. The idea of this paper is composed of two parts:

• The out of equilibrium dynamics under these circumstances is a very specific one [4–9]. At given times one may define a temperature with a thermodynamic meaning [10], it is the same for all observables. Different temperatures are possible, but in diferent 'scales', a notion one has to define. We refer to this situation as 'multithermalization' [11,12]. Rather unexpectedly, the temperatures involved in the slow dynamics coincide with a series of parameters computed for equilibrium in the Parisi scheme. This may be argued on the basis of a strategy devised by Franz, Mézard, Parisi and Peliti (FMPP) [13,14] years ago, in a remarkable work to which we will refer throughout this paper.

• Consider two such systems brought into weak contact from the beginning, for example two different lattice models, coupled locally so as to obtain a single model with two sublattices. Assume that at the same times the separate systems have non-coincident temperatures. Does this mean that the coupled system, for which there is no distinction between observables of one or the other system, violates the multithermalization scenario - and, a fortiori, the Parisi scheme? If this were so, both would be fragile to the point of irrelevance. The answer is surprising: the timescales of the systems rearrange so that different temperatures happen at different scales: thus, the combined system conforms to the scenario. The fact that this needs to happen *for infinitesimal coupling* means that the system needs to be 'soft' with respect to time-rearrangements of each temperature separately: in other words, it has to have independent time-reparametrization invariances in the slow dynamics limit. Such invariances where first described by Sompolinsky and Zippelius [5,6] some forty years ago, and have recently had a crucial role in the interpretation of the SYK model [15] as a toy model of holography [16,17].

Now, it is quite natural to assume that time-reparametrization invariances, an independent one for each temperature, will only be possible if such temperatures happen at widely separated timescales, because then one may 'move each timescale around' without changing their mutual interaction: overlapping timescales would make this invariance unlikely. This hierarchy in times, already found in mean-field problems, seems then like a general necessity for having a unique temperature for all observables at given times, and ultimately for the correspondence between dynamics and Parisi scheme. The reader who is convinced by this heuristic argument may skip sections III and IV. In those sections, we extend the procedure of FMPP to confirm, within their framework, that separation of timescales indeed is necessary for the agreement between dynamics and Parisi scheme.

Time reparametrizations and the unambiguous definition of timescales, when we are dealing with observables that depend on two or more times, require some clarifying definitions, a large part of which have been already discussed in the past. Most importantly, it is convenient to separate those quantities that are reparametrization-invariant from the reparametrizations themselves, a procedure that may even be implemented experimentally: see [18–23].

## 1.1 Equilibrium

The Parisi construction [1] involves the computation of the Boltzmann-Gibbs distribution, averaged over quenched disorder. The measure is given by an infinite set of *pure states*, each state $\alpha$ a set of configurations– just like the positive and negative magnetization distributions

in a ferromagnet – inside which a variable $s_i$ has expectation value $\langle s_i \rangle_\alpha$ (e.g. in a ferromagnet, $\langle s_i \rangle_\pm = \pm m$). The overlap between two states is, for example for a spin system $q^J_{\alpha\beta} = \frac{1}{N} \sum_i \langle s_i \rangle_\alpha \langle s_i \rangle_\beta$, where the supraindex $J$ signifies that we have not yet averaged over disorder. Once we do, we obtain $q_{\alpha\beta} = \overline{q^J_{\alpha\beta}}$. A histogram of the $q^J_{\alpha\beta}$ for a given disorder is mostly dominated by a few spikes, while the average histogram for $q_{\alpha\beta}$ is the Parisi function $P(q)$, a direct product of the formalism. The same information is contained in the primitive $x(q)$, such that $\frac{dx}{dq} = P(q)$. The other hallmark of the Parisi ansatz is the 'ultrametricity' property: for any three states at mutual overlaps $q_{12}, q_{23}, q_{31}$ the two smallest overlaps are equal (all triangles are isosceles): $q_{13} = \min(q_{12}; q_{23})$

In fact, the ultrametric solution may be proven [24] from two hypotheses: *i)* stochastic stability (see Ref. [25]): the solution keeps its form under small random perturbations, and *ii)* overlap equivalence [24, 26]: all the mutual information about a pair of equilibrium configurations is encoded in their mutual distance or overlap. In other words, we may always write the correlation of an observable in two states as a function of that of another observable in the same states: $\bar{q}_{ab} = g(q_{ab})$ where $g$ is a smooth function. In what follows, when we refer to 'Parisi scheme', we consider it assuming these two properties.

## 1.2 Dynamics

In the dynamic approach we have an evolving system:

$$-m_i \ddot{s}_i - \frac{\partial V(\mathbf{s})}{\partial s_i} = \underbrace{\Gamma_0 \dot{s}_i - \eta_i}_{bath}, \tag{1}$$

where $\eta_i$ are uncorrelated Gaussian white noises with variance $2\Gamma_0 T$ and $\Gamma_0$ is the strength of the coupling to the 'white' bath. This is guaranteed to reach eventually equilibrium, although in the systems that concern us, in times that may diverge with $N$.

We are interested in various correlation $C_{AB}(t, t')$ and response functions $R_{AB}(t, t')$ (here, and in what follows, always $t \geq t'$), the average response of $A$ at time $t$ to a kick of $B$ at time $t'$. From here, we read the correlations and response functions:

$$C_{ij}(t, t') = \langle s_i(t) s_j(t') \rangle \qquad ; \qquad R_{ij}(t, t') = \left\langle \frac{\delta s_i(t)}{\delta h_j(t')} \right\rangle, \tag{2}$$

where $h_i$ is a field conjugate to $s_i$. We shall often use:

$$C(t, t') = \frac{1}{N} \sum_i C_{ii}(t, t') \qquad ; \qquad R(t, t') = \frac{1}{N} \sum_i R_{ii}(t, t'), \tag{3}$$

$$\chi(t, t') = \theta(t - t') \int_{-\infty}^{t'} dt'' \, R(t, t''), \tag{4}$$

(note that the definition with these limits of integration is rather unusual) and the symmetrized version

$$\chi_s(t, t') = \chi(t, t') + \chi(t', t). \tag{5}$$

In the spirit of the fluctuation-dissipation theorem, we will define effective temperatures [10] as:

$$T_{AB}(t, t') R_{AB}(t, t') = \frac{\partial C_{AB}(t, t')}{\partial t'} \qquad ; \qquad T_{AB}(t, t') = \frac{T}{X_{AB}(t, t')}. \tag{6}$$

In equilibrium $X = 1$ and where $T_{AB}(t, t') = T$, the bath's temperature.

When there is time-translational invariance (TTI),

$$\chi(t-t') = \chi(\tau) = \int_\tau^\infty d\tau' \, R(\tau') \qquad ; \qquad R(\tau) = -\chi'(\tau) \; for \; (\tau > 0) \tag{7}$$

and a short calculation gives for the Fourier transforms:

$$i\omega \hat{\chi}_s(\omega) = [\hat{R}(\omega) - \hat{R}^*(\omega)]. \tag{8}$$

We may consider many different settings for dynamics, but here we shall only be concerned with the limit of slow dynamics, which may be achieved at least in three ways:

- *Aging [27]:* We quench the system from a high to a low temperature, at which the equilibration time is infinite. The system 'ages': it evolves slower and slower as the time since the quench elapses. The two-point functions never fully become a function of time-differences. The large parameter is the smallest 'waiting' time since the quench $t' = t_w$, that modulates the decay at $\tau = t - t'$. A typical example is $C(t,t') = C\left(\frac{\tau}{t_w}\right)$.

- *Driven system [2,3]* When the system is subjected to forces non deriving from a potential – shear, for example – it is an experimental fact that aging is interrupted, in the sense that all functions become time-translational invariant, but slow. Their timescale of the decay of correlation then is controlled by the driving rate $\sigma$, the slower the weaker the drive: $C(t-t') = C(\tau\sigma)$.

- *Time-dependent disorder [4].* Another way to make a system with disorder time-translational invariant is to change the disorder slowly : the small parameter is the timescale $\tau_0$ of change of disorder: $C(t,t') = C\left(\frac{\tau}{\tau_0}\right)$. The reason is simple: the system optimises with a constantly changing target.

In the case of mean-field glasses, we know that the three situations above correspond, in the limit of slow dynamics, to different time-reparametrizations of the same solution. We shall discuss below the condition for this being the case in finite-dimensions. In what follows, we will refer briefly as *'asymptotic'* to the limit of either long waiting times, small shear strains or slow variation of parameters, always taken *after* the thermodynamic limit.

## 2 The framework

### 2.1 Factoring out time - general kinematic constraints.

Although one may ask about the time-dependence of any quantity, it turns out that there is a particularly significant sub-ensemble of dynamic quantities: those where time is factored out [8], and are thus invariant under reparametrizations $t \to h(t)$. This is achieved, as we shall see, by using a single correlation as a 'clock':

- Given any dynamic parameter $X(t,t')$ define for large times $X(t,t') \to X[C(t,t')] = \lim_{t'\to\infty} X[C(t,t'),t']$. We shall focus on cases in which this limit is non-trivial. This also implies that the integrated response becomes a function of the correlation: $\chi(t,t') \to \chi[C](t,t')$.

- given three long, successive times $t_1 < t_2 < t_3$, and the corresponding correlations $C_{21}, C_{32}, C_{31}$, define for large times $C_{31} = f(C_{21}; C_{32}) = \lim_{t_1\to\infty} f(C_{21}; C_{32}, t_1)$, a 'triangle relation'. It is easy to show that $f(a,b)$ is an associative function of $a$ and $b$ (see construction Fig. 1). Similarly for the remanent magnetizations $\chi(C_{31}) = \tilde{f}[\chi(C_{21}); \chi(C_{32})]$, i.e. the triangle relations $\tilde{f}$ and $f$ are isomorphic.

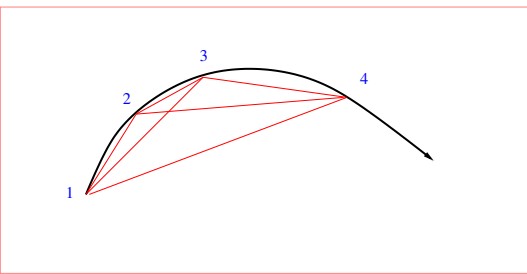

Figure 1: The proof that $C_{41} = f[C_{43}, f(C_{32}, C_{21})] = f[f(C_{43}, C_{32}), C_{21}]$: the function $f$ is associative.

- Given any two correlations of the system $\bar{C}(t, t')$ and $C(t, t')$ we write, again in the large times limit, $\bar{C} \to g(C)$ for some $g$.

The function $f$ is associative and may be classified as such: *a purely 'kinematic' construction, independent of the dynamics*. It is shown in [8] that there are 'skeleton values' $q_r$ of $C$ which delimit *correlation scales* $\mathcal{S}_C$, such that:

- If $C_{21}$ and $C_{32}$ are both in the same scale, then $f(C_{21}, C_{32}) = g^{-1}[g(C_{21}) + g(C_{32})]$, i.e. $f$ is isomorphic to the sum (or the product). The function $g$ is a different one for each interval.

- If $C_{21}$ and $C_{32}$ are in the different scales, then $f(C_{21}, C_{32}) = \min[g(C_{21}), g(C_{32})]$. This means that the relaxations in different scales take place in very different timescales, so that the time for relaxing within one scale is negligible with respect to the other.

- From this it follows that there is always a time-reparametrization that makes the correlation within a scale time-translational invariant, that is: $C_{21} = C(t_2 - t_1)$. For example, if a correlation is of the form $C\left(\frac{t'}{t}\right)$, then $h(t) = \ln t$ is such a mapping. Note that if there is more than one scale, the times are reparametrized differently for the correlations in each scale. (examples below).

## 2.2 Three examples

**Two scales**

This is the most usual case. An example is when the correlation $0 < C \le 1$ and there is a value $q$ such that for the interval $q \le C \le 1$ the correlation is much faster than for the interval $0 \le C < q$. We have, for example:

- For a stationary case $C(t - t') = (1 - q)\bar{A}(t - t') + q\,\bar{B}\left(\frac{t - t'}{H(\tau_0)}\right)$, where $H$ is a growing function of $\tau_0$.

- For an aging case $t > t'$:
  $C(t, t') = (1 - q)A(t - t') + q\,B\left(\frac{L(t')}{L(t)}\right) = (1 - q)A(t - t') + q\,B\left(e^{h(t) - h(t')}\right),$

where $(A, B, \bar{A}, \bar{B})$ are functions decreasing from one to zero as their argument goes from zero to infinity. We have put $h(t) = \ln L(t)$ to emphasize that the form may be brought into a time-translational invariant form via a reparametrization.

The stationary case has separated timescales as $\tau_0 \to \infty$, and the aging one at long times $t$. The aging form is in particular the one of domain growth, where the fast part $\bar{A}(t - t')$ is

the relaxation within domains, and the slow part is a function of the domain length $L(t)$. It is easy to see that in this limit $f$ is isomorphic to the addition within each scale $(0-q)$ and $(q-1)$, and is the function min for correlations in different scales.

**Three scales**

Again, the correlation $0 < C \leq 1$ and there two values $q_0$ and $q_1$ such that for the interval $q_1 \leq C \leq 1$ the correlation is much faster than for the interval $q_0 \leq C < q_1$, itself much faster than $0 \leq C < q_0$ We have, for example, for the stationary state:

- $C(t - t') = (1 - q_1)\bar{A}(t - t') + (q_1 - q_0)\bar{B}\left(\frac{t-t'}{H(\tau_0)}\right) + q_0 \bar{\bar{B}}\left(\frac{t-t'}{\bar{H}(\tau_0)}\right),$

where $\bar{\bar{B}}$ is also decreasing from one to zero as their argument goes from zero to infinity. The timescales are nested as $\tau_0 \to \infty$: $\bar{H}(\tau_0) \gg H(\tau_0) \gg 1$.

The function $f$ is the function min for correlations in any two different scales.

**A continuum of scales**

An important case is when there is a dense set of values of correlation in which for all values of correlation

$$f(C_{21}, C_{32}) = \min[g(C_{21}), g(C_{32})] \tag{9}$$

holds. An example is:

$$C(t, t') = \mathcal{C}\left(\frac{\ln(t - t' + t_0)}{\ln \tau_0}\right), \tag{10}$$

where $t_0$ is a constant. This form satisfies (9) when $\tau_0 \to \infty$.[1] It may be viewed as an infinite superposition of scales, e.g:

$$C(\tau) = \int d\nu \, [\tau_0]^\nu \, e^{-\tau_0^{-\nu}(\tau + t_0)} \, \mathcal{C}(-\nu) \propto \mathcal{C}\left(\frac{\ln(\tau + t_0)}{\ln \tau_0}\right), \tag{11}$$

where we have evaluated the integral by saddle-point over $\nu$.

## 2.3  Time-reparametrizations

In the sections above, we have written everything using one particular correlation as a 'clock', time-dependencies are mediated by that correlation. Note that this is also possible with higher order correlations. We should now define clearly which time-reparametrizations we shall consider. The answer is simple: *those that preserve the triangle relations*. For example, if the system has two scales, it is easy to see that a possibility is:

$$\{t, t'\} \to \{t, t'\} \text{ for } q \leq C \leq 1 \text{ and } \{t, t'\} \to \{h(t), h(t')\} \text{ for } 0 \leq C \leq q. \tag{12}$$

Note that *i)* we have two different reparametrizations for the two scales, and *ii)* we do not reparametrize the fast (ultraviolet) scale, because it is the one for which reparametrization invariance of the action does not hold – because time-derivatives are not negligible there.

For two or more scales, this is easily generalizable to

$$\{t, t'\} \to \{h_\mathcal{S}(t), h_\mathcal{S}(t')\}, \tag{13}$$

---

[1] Note however that, confusingly, $C(t, t') = \mathcal{C}\left(\frac{\ln t'}{\ln t}\right)$ is only *one* scale!

where the reparametrization depends on the scale $\mathcal{S}$ to which $C(t, t')$ belongs.

Let us note that this may allow more freedom than reparametrizations found in in models such as SYK, because each scale is reparametrized separately. This is most clearly seen in the case in which there is a continuum of scales, for example Eq (10). In that case, $f$ is invariant with respect to reparametrization of *time-differences* $(t - t') \rightarrow H(t - t')$ for any smooth $H$, something that does not happen for discrete scales.

**An example of factoring times away:**

For a triangle of correlations $[C(t_{int}, t_{min}), C(t_{max}, t_{int}), C(t_{max}, t_{min})]$, we define, asymptotically [8]:

$$
\begin{align}
C(t_{max}, t_{min}) &\rightarrow f[C(t_{int}, t_{min}), C(t_{max}, t_{int})], \tag{14} \\
C(t_{max}, t_{int}) &\rightarrow \bar{f}[C(t_{int}, t_{min}), C(t_{max}, t_{min})] \geq C(t_{max}, t_{min}), \tag{15} \\
C(t_{int}, t_{min}) &\rightarrow \bar{\bar{f}}[C(t_{max}, t_{int}), C(t_{max}, t_{min})] \geq C(t_{max}, t_{min}), \tag{16}
\end{align}
$$

when the $f = \min$ we have:

$$
\begin{align}
C(t_{max}, t_{int}) &\rightarrow C(t_{max}, t_{min}) \quad \text{if} \quad C(t_{max}, t_{min}) \leq C(t_{int}, t_{min}), \tag{17} \\
C(t_{int}, t_{min}) &\rightarrow C(t_{max}, t_{min}) \quad \text{if} \quad C(t_{max,tmin}) \leq C(t_{max}, t_{int}). \tag{18}
\end{align}
$$

Let us see how we use these in an example. In computing the dynamic diagrams we will meet later, we shall need to calculate convolutions such as:

$$
I(t, t') = \int_{-\infty}^{t'} C(t, t'') R(t', t'') \, dt''. \tag{19}
$$

Introducing the definition of $X$:

$$
\int_{-\infty}^{t'} C(t, t'') X(t', t'') \frac{\partial C}{\partial t''}(t', t'') \, dt''. \tag{20}
$$

Now we may factor times away:

$$
\bar{I}(C) = \int_0^C C' X(\bar{f}(C')) \frac{d\bar{f}(C', C)}{dC'} \, dC'. \tag{21}
$$

It turns out that all integrals coming from diagrammatic expansions may be treated this way.

## 2.4 Dynamic multithermalization properties

The properties above do not really use any property of the dynamics, except that it should *have* a slow regime. The one we discuss here instead implies a definite assumption on the dynamics. It is inspired in the mean-field solution.

**Thermalization as a residual symmetry**

Let us construct the path-integral generator [28–30] associated to the equation of motion (1). Introducing a Fourier variable $\hat{s}_i$ and integrating over noise, we get:

$$
Z = \int D[s] D[\hat{s}] \exp \left\{ \int dt \sum_i \hat{s}_i \left( -m_i \ddot{s}_i - \frac{\partial V(\mathbf{s})}{\partial s_i} - \Gamma_0 (\dot{s}_i + T \hat{s}_i) \right) \right\}, \tag{22}
$$

in the Ito convention, which means the determinant term is absent. From here, we read the correlations and response functions:

$$C_{ij}(t,t') = \langle s_i(t)s_j(t')\rangle \qquad ; \qquad R_{ij}(t,t') = \langle s_i(t)\hat{s}_j(t')\rangle , \qquad (23)$$

where $h_i$ is a field conjugate to $s_i$. Detailed balance implies that the time-reversal symmetry

$$s_i(t) \to s_i(-t) \qquad ; \qquad \hat{s}_i(t) \to \hat{s}_i(-t) + \beta \dot{s}_i(-t) \qquad (24)$$

leaves the integral invariant up to a boundary term in time. This is the time-reversal detailed-balance property. In particular, if the symmetry is unbroken this implies that all functions depend on time-differences and:

$$C_{AB}(t-t') = C_{BA}(t-t') \qquad ; \qquad \chi_{AB}(t-t') = \beta C_{AB}(t-t') - \chi_{AB}(t'-t). \qquad (25)$$

If the bath is absent $\Gamma = 0$, and this symmetry holds for any $\beta$ corresponding to the energy of the initial condition. The presence of the bath breaks the larger symmetry to a subgroup [31, 32], given by the $\beta$ of the bath. As we shall see below, this may happen spontaneously in each timescale, and with a different $\beta$: we know for sure that this scenario is valid within mean-field, and we shall discuss below what are the implications of it holding for finite-dimensional systems.

**Multi-thermalization as a symmetry-breaking scheme**

We have seen above that within a correlation scale, the correlation and response functions are such that the 'triangle relation' is smooth, and hence isomorphic to the sum (or the product)

$$g(C_{31}) = g(C_{32}) + g(C_{21}) \qquad ; \qquad \chi(t,t') \to \chi[C(t,t')]. \qquad (26)$$

This implies that they may be written as:

$$C(t,t') \to \mathcal{C}[h(t) - h(t')] = \mathcal{C}[h-h'] \qquad ; \qquad \chi(t,t') \to \mathcal{K}[h(t)-h(t')] = \mathcal{K}[h-h'] \qquad (27)$$

and similarly for all correlations and response functions of any number of times. For example, in an aging situation, $h(t) \sim \ln t$ yields a $\left(\frac{t'}{t}\right)$-dependence, an ansatz often made.

If a timescale is isolated we get to a point at which the 'kinetic' terms may be neglected. Having assumed that within a scale all functions depend on differences of $h$'s, in terms of these we have a corresponding 'time' reversal symmetry $h \to -h$ associated to some $\tilde{\beta}$. This implies:

$$\mathcal{C}_{AB}(h-h') = \mathcal{C}_{AB}(h'-h) \qquad ; \qquad \mathcal{K}_{AB}(h-h') = \tilde{\beta}\mathcal{C}_{AB}(h-h') - \mathcal{K}_{AB}(h'-h). \qquad (28)$$

All in all, we have these symmetries parametrized by $\beta_{\mathcal{S}}$ in each timescale $\mathcal{S}$, and:

$$T(C) \neq T(C') \qquad \Rightarrow \qquad f(C,C') = \min(C,C'). \qquad (29)$$

In particular, when there is a continuum of timescales, there is in general a continuum of temperatures. There is a well-defined temperature for all observables within this scale, plus Onsager reciprocity.

This is then a symmetry breaking to a smaller group scheme [11, 12], labeled by the temperatures of each scale: as such it is consistent, but of course need not be the correct solution of a given problem. The Parisi scheme for statics is also a symmetry breaking scheme into subgroups [1](of the permutation group of a noninteger number of elements). One might suspect that there is a correspondence between the two schemes. Both are known to apply to mean-field statics and dynamics. In what follows we shall argue, within the assumption of stochastic stability [25] (w.r.t long-range perturbations), that this correspondence is a necessity in finite dimensions.

# 3 Connections between dynamic and Parisi scheme

## 3.1 A first, formal bridge between dynamic and static (replica) and calculations

This formal bridge has been known for a long time [33], and sometimes used for calculations. As is well known, a path integral like (22) may be written in a compact form in therms of the 'superspace' variables:

$$\mathbf{s}_i(1) = s_i(t) + \theta \bar{\eta} + \bar{\theta} \eta + \bar{\theta} \theta \hat{s}(t), \tag{30}$$

where $\theta_a$, $\bar{\theta}_a$ are Grassmann variables, and we denote the full set of coordinates in a compact form as $1 = t_1 \theta_1 \bar{\theta}_1$, $d1 = dt_1 d\theta_1 d\bar{\theta}_1$, etc. $\bar{\eta}, \eta$ are fermion variables that play no role here, and will be hence ommited. This notation brings the replica and dynamic treatment into formally very close contact, with one-to-one (topological) correspondence between diagrams. We write Eq (22) as:

$$\int D\mathbf{s} \exp\left\{ \int d1 [K(\mathbf{s}) - V(\mathbf{s})] \right\}, \tag{31}$$

where

$$K(\mathbf{s}) = \sum_i \frac{\partial \mathbf{s}_i}{\partial \theta} \left( \frac{\partial \mathbf{s}_i}{\partial \bar{\theta}} - \theta \frac{\partial \mathbf{s}_i}{\partial t} \right) - \frac{\partial^2 \mathbf{s}_i}{\partial t^2} \tag{32}$$

is a 'kinetic' term which contains the time-derivatives, which will be neglected in the slow-dynamics regimes.

We encode the correlations and (causal) responses in the 'superspace' order parameter (see [33]):

$$Q_{ij}(1,2) = C_{ij}(t_1, t_2) + (\bar{\theta}_2 - \bar{\theta}_1) \left[ \theta_2\, R_{ij}(t_1, t_2) - \theta_1\, R_{ji}(t_2, t_1) \right] \tag{33}$$

and similarly

$$Q(1,2) = C(t_1, t_2) + (\bar{\theta}_2 - \bar{\theta}_1) \left[ \theta_2\, R(t_1, t_2) - \theta_1\, R(t_2, t_1) \right], \tag{34}$$

which corresponds to the matrix

$$Q(t, t') = \begin{bmatrix} R(t, t') & C(t, t') \\ 0 & R(t', t) \end{bmatrix}. \tag{35}$$

As we shall see below, we will be led, in this notation, to topologically equal diagrams for replicas and dynamics, with the identifications

$$\sum_{\alpha=1}^{n} \rightarrow \int d1, \tag{36}$$

our diagrams will have vertices at supertimes $1 = (t, \bar{\theta}, \theta)$ (replica $\alpha$, respectively) and lines given by $Q(1,2)$ (respectively $Q_{\alpha\beta}$). One of the lines will be integrated with a generating variable $d1 \rightarrow d1 j(1)$, (respectively $\sum_{\alpha} \rightarrow \sum_{\alpha} j_{\alpha}$, where $j$ are the arguments of the generating functions:

$$j(\alpha) = 1 + j\delta_{1\alpha} \rightarrow j(1) = 1 + j\delta(t_1 - t_0)\bar{\theta}_1\theta_1. \tag{37}$$

We shall see a few examples of this below. The fact that the diagrams have the same form does not automatically mean that their actual values are the same. It has been long known [33] that, in the case in which there is a single temperature per timescale, then the results of dynamic diagrams and Parisi-ansatz replica ones are indeed the same diagram by diagram, the question that we shall address in what follows is whether timescale-separation is also necessary for this to happen.

# 4 Properties derived from stochastic stability

We shall assume that the properties of the system are unchanged when perturbed by random, weak *but long-range* interactions: *stochastic stability*. Under this assumption we shall show that the multithermalization and Parisi schemes imply one another, for finite-dimensional systems.

## 4.1 Same temperatures for all observables implies separation of timescales

Let us show first that the only way that such a system has the same $T(t, t')$ for all observables at the same $(t, t')$ is that there is only one temperature for all $(t, t')$ associated to a correlation scale. In other words, non-constant temperature within a timescale implies that different observables have different temperatures at the same times. Later on, we will see that this implies that there is no *overlap equivalence*, at the static level.

Let us first consider a lattice system, which we divide in four sublattices. whose components we shall call $s_i^{(1)}$, $s_i^{(2)}$ and $s_i^{(3)}$ and $s_i^{(4)}$. Adding to the energy a term

$$
\begin{aligned}
S = \ & \frac{1}{2} \sum_{ij} \left( h_{ij}^{(1)} \right)^2 + \frac{1}{2} \sum_{ij} \left( h_{ij}^{(2)} \right)^2 + \frac{1}{2} \sum_{ij} \left( h_{ij}^{(3)} \right)^2 + \frac{1}{2} \sum_{ij} \left( h_{ij}^{(4)} \right)^2 \\
& + \frac{\gamma}{N} \sum_{ij} \left( h_{ij}^{(1)} h_{jk}^{(2)} s_k^{(2)} s_i^{(1)} + h_{ij}^{(2)} h_{jk}^{(3)} s_i^{(2)} s_k^{(3)} + h_{ij}^{(3)} h_{jk}^{(4)} s_i^{(3)} s_k^{(4)} \right),
\end{aligned}
\tag{38}
$$

with the $h_{ij}^{(\ell)}$ random Gaussian variables. We wish to compute the following correlation and its associated response:

$$
C^{(4)}(t, t') = \frac{1}{N^2} \sum_{ijk} h_{ij}^{(1)} h_{jk}^{(4)} \left\langle s_i^{(1)}(t) s_k^{(3)}(t') \right\rangle_\gamma ; \quad R^{(4)}(t, t') = \frac{1}{N^2} \sum_{ijk} h_{ij}^{(1)} h_{jk}^{(4)} \left\langle \frac{\delta s_i^{(1)}(t)}{\delta \eta_k^{(3)}(t')} \right\rangle_\gamma .
\tag{39}
$$

These may be encoded in a superspace order parameter $Q^{(4)}(1, 2)$, or in its matrix version:

$$
Q^{(4)}(t, t') = \begin{bmatrix} R^{(4)}(t, t') & C^{(4)}(t, t') \\ 0 & R^{(4)}(t', t) \end{bmatrix} = \gamma^3 [Q \otimes Q \otimes Q \otimes Q](t, t'),
\tag{40}
$$

where $\otimes$ stands for convolution and matrix product. Or, equivalently, in superspace notation:

$$
Q^4(1, 2) = \gamma^3 [Q]^4(1, 2).
\tag{41}
$$

Note that this is a convolution power. As we have seen in Section II, we may always assume, by reparametrizing times within a timescale, that the functions are time-translational invariant, and we may use Fourier transforms:

$$
Q(t - t') = \begin{bmatrix} R(t - t') & C(t - t') \\ 0 & R(t' - t) \end{bmatrix} \to \hat{Q}(\omega) = \begin{bmatrix} \hat{R}(\omega) & \hat{C}(\omega) \\ 0 & \hat{R}^*(\omega) \end{bmatrix}.
\tag{42}
$$

The generalization to $n$ sublattices is obvious:

$$
\hat{Q}^{(n)}(\omega) = \begin{bmatrix} \hat{R}^{(n)}(\omega) & \hat{C}^{(n)}(\omega) \\ 0 & \hat{R}^{(n)*}(\omega) \end{bmatrix}.
\tag{43}
$$

We have thus constructed a whole set of pairs a set $R^{(n)}$, $C^{(n)}$ starting from $C$, $R$: is it possible that they are related by the same $T_{eff}$ at equal times? It turns out that it is much easier to answer this question by comparing $n = 1$ with large $n$, this result on its own will tell us that a necessary condition is to have timescale separation.

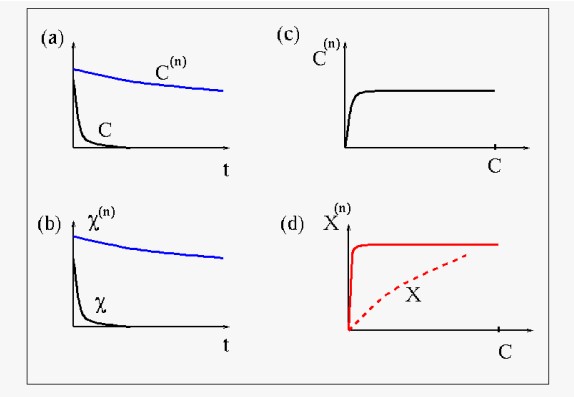

Figure 2: Illustration of the fact that $C^{(n)}$ and $\chi^{(n)}$ become broader and broader with $n$, $X^{(n)}$ and essentially flat. Within that range, if $X(C)$ is not a constant, then $X^{(n)}(t, t') \neq X(t, t')$.

A short calculation gives

$$\hat{R}^{(n)} = \hat{R}^n \qquad ; \qquad \hat{C}^{(n)}(\omega) = \frac{\hat{R}^{(n)}(\omega) - \hat{R}^{(n)*}(\omega)}{\hat{R}(\omega) - \hat{R}^*(\omega)} \, \hat{C}(\omega), \tag{44}$$

this may also be written as:

$$\frac{\hat{C}^{(n)}(\omega)}{\hat{\chi}_s^{(n)}(\omega)} = \frac{\hat{C}(\omega)}{\hat{\chi}_s(\omega)}. \tag{45}$$

If we consider a large value of $n$, then $\hat{C}^{(n)}(\omega)$ and $\hat{\chi}_s^{(n)}(\omega)$ will be peaked around some value of $\omega$ dominated by the maximum of $R(\omega)$, which for a strongly dissipative system we expect to be zero. Then we may write:

$$\frac{\hat{C}^{(n)}(\omega)}{\hat{\chi}_s^{(n)}(\omega)} \sim \frac{\hat{C}(0)}{\hat{\chi}_s(0)} = \bar{T}. \tag{46}$$

From which we immediately see that $R^{(n)}(\tau), C^{(n)}(\tau)$ satisfy fluctuation dissipation with the average temperature. Now, if $R(\tau), C(\tau)$ do not have a single temperature, then at equal times both pairs of observables have different temperatures, because $C^{(n)}$ and $\chi^{(n)}$ become broader and broader with $n$, $X^{(n)}$ as is depicted in Figure 2.

## 4.2 Relation between statics and dynamics in finite dimensions

In FMPP it is shown that, under certain assumptions, the dynamic $X(C)$ and the equilibrium counterpart $x(q)$ coincide for finite-dimensional systems. This is at first sight very strange, since it concerns a relation between two different kinds of objects that are relevant in completely different time regimes (in and out of equilibrium, respectively), and happen in different regions of phase-space. This section is mostly a review of their results.

The Parisi scheme gives us the Parisi order parameter $P(q)$, the probability of overlaps of states, averaged over disorder. We shall sometimes need to distinguish the values of $q$ where $P(q)$ is nonzero: we shall for brevity call them 'skeleton' values. This distinction becomes important when we consider the next defining feature of the Parisi construction: the structure of triangles determined by three states $(q_{13}; q_{12}; q_{23})$. By its very definition, this may only concern skeleton values of the $q$. The natural next question is what becomes of the ultrametricity property of statics: is there any relation between the dynamic triangle relation $C_{13} \rightarrow f(C_{12}; C_{23})$

and ultrametricity of equilibrium states $q_{13} \to \min(q_{12}; q_{23})$? *Clearly, the second $f$ concerns all values of C, while the static ones only the skeleton values, so if there is a correspondence it has to be for the skeleton values only.*

Dynamically, if we had $C_{13} \to \min(C_{12}; C_{23})$ it would mean that we have hierarchically organized timescales. For example, if the system is TTI then one of the two time differences $t_2 - t_1$ and $t_3 - t_2$ is negligible compared to the other, so correlations make their steps of decay on widely separated times. Franz et al asked whether Parisi scheme implied the existence of widely separated timescales. Their conclusion was that timescale separation is sufficient for having a Parisi scheme, but, though considering it plausible, left open the question as to whether it was also a *necessary* one. In this paper it is shown that their same scheme also shows that indeed this is so: widely separated timescales are indeed implied by the Parisi scheme, at least such as we know it (i.e. with overlap equivalence and stochastic stability). This closes the circle: for finite dimensional systems the Parisi scheme is included in the dynamic multi-thermalization one, and it allows to compute some of its dynamic relations for which time has been factored away.

This connection between widely separated timescales and the Parisi scheme will lead us to the main point of this paper, the question of time-reparametrization invariances: we shall show that these are crucial for the consistency of the scheme, since they allow two systems brought into contact to 'adjust their timescales' so that different effective temperatures match at each scale.

**The basic argument**

The idea in FMPP is to compute the generalized susceptibilities defined as follows [7]: one adds a perturbation of the form

$$H \to H + \left\{ \frac{\gamma}{N^{(p-1)/2}} \sum h_{i_1,...,i_p} s_{i_1} ... s_{i_p} + \sum h^2_{i_1,...,i_p} \right\}, \tag{47}$$

where the $h_{i_1,...,i_p}$ are gaussian iid random numbers, and computes the susceptibility

$$I^{(p)} = \frac{\gamma}{N^p} \frac{\partial}{\partial \gamma} \left\langle \sum h_{i_1,...,i_p} s_{i_1} ... s_{i_p} \right\rangle. \tag{48}$$

In equilibrium, and asymptotically for dynamics, we have

$$I^{(p)}_{equil} = \int x(q) dq^p \qquad ; \qquad I^{(p)}_{dyn} = \int X(C) dC^p. \tag{49}$$

These correspond, in the notation introduced above, to:

$$I^{(p)}_{equil} = \frac{d}{dj} \sum_{\alpha\beta} Q^{\bullet p}_{\alpha\beta} \, j(\alpha) \bigg|_{j=0} \qquad ; \qquad I^{(p)}_{dyn} = \frac{d}{dj} \int d1 \, d2 \, Q^{\bullet p}(1,2) j(1) \bigg|_{j=0}, \tag{50}$$

where $A^{\bullet p}$ is the matrix each of whose elements is the $p$-th power of that of the matrix $A$, a Hadamard (element-by-element) power. This corresponds to the left diagram in Fig. 3. Now, if one can argue that these susceptibilities are (to leading order in $N$ and long times taken after $N \to \infty$) equal for all $p$, then one concludes that $x(q)$ and $X(C)$ are the same functions. The argument to show this is that, since these susceptibilities may be obtained from a second derivative of a free-energy with respect to sources , equality of energy *densities* between dynamics and equilibrium (again, to leading order in $N$ and long times, and for all small perturbations) implies equality of susceptibilities. Franz et al used the standard nucleation

reasoning forbidding stable states with higher free-energy density, *valid in finite dimensions,* to argue this. In order to complete the argument, they need to get around an obstacle: a term like $h_{i_1,...,i_p} s_{i_1}...s_{i_p}$ is long-range, and the nucleation argument would not, in principle, apply. Their clever trick is to consider the $d$-dimensional lattice and mentally fold it $p-1$ times, so as to make $(i_1,...,i_p)$ contiguous. The resulting $(p-1)$-layer system is then short-range, and we may apply the nucleation argument. This applies for a single term, and one should then argue that it also does for the sum of them all. For this step one needs that the limit of small perturbation and thermodynamic limit commute, and this is where some form of stochastic stability is required.

It is easy to see that the whole argument of FMPP extends naturally to the case in which disorder changes slowly, because by making the timescale long enough all nucleations may take place. The same is true for the weak-shear limit.

Now, to prove the correspondence of FMPP in a compact form, we write:

$$Z_h(\gamma) = \Sigma_{\alpha,i}\, e^{S_0 + S(h, s_i^\alpha)}, \tag{51}$$

where

$$
\begin{aligned}
S &= \left\{ \frac{\gamma}{N^{(p-1)/2}} \sum_{i_1,...,i_p} h_{i_1,...,i_p} \sum_a j(a) s_{i_1}^a ... s_{i_p}^a + \frac{1}{2} \sum_{i_1,...,i_p} h_{i_1,...,i_p}^2 \right\} \\
&= \left\{ \frac{\gamma}{N^{(p-1)/2}} \sum_{i_1,...,i_p} h_{i_1,...,i_p} \mathbf{t}_{i_1,...,i_p} + \frac{1}{2} \sum_{i_1,...,i_p} h_{i_1,...,i_p}^2 \right\},
\end{aligned}
\tag{52}
$$

where here $t_{i_1,...,i_p} \equiv \sum_a j(a) s_{i_1}^a ... s_{i_p}^a$ and $j(a) = 1 + j\delta(t_1 - t_o)\delta_{ab}$.

Similarly, dynamically we have:

$$Z_h(\gamma) = \Sigma_{\alpha,i}\, e^{S_0 + S(h, \mathbf{s}_i)}, \tag{53}$$

where

$$
\begin{aligned}
S &= \left\{ \frac{\gamma}{N^{(p-1)/2}} \sum_{i_1,...,i_p} h_{i_1,...,i_p} \int d1\, j(1) \mathbf{s}_{i_1} ... \mathbf{s}_{i_p}(1) + \frac{1}{2} \sum_{i_1,...,i_p} h_{i_1,...,i_p}^2 \right\} \\
&= \left\{ \frac{\gamma}{N^{(p-1)/2}} \sum_{i_1,...,i_p} h_{i_1,...,i_p} \mathbf{t}_{i_1,...,i_p} + \frac{1}{2} \sum_{i_1,...,i_p} h_{i_1,...,i_p}^2 \right\},
\end{aligned}
\tag{54}
$$

where $\mathbf{t}_{i_1,...,i_p} \equiv \int d1\, j(1) \mathbf{s}_{i_1} ... \mathbf{s}_{i_p}(1)$ and $j(1) = 1 + j\delta(t_1 - t_o)\bar{\theta}_1 \theta_1$. Integrating over the $h$'s we get the diagram of the left of in Fig 3.

**Generator function Lego**

It is natural to extend this to more general

$$H \rightarrow H + \left\{ \frac{\gamma}{N^{(p-1)/2}} \sum h_{i_1,...,i_p} s_{i_1}...s_{i_p} + \sum h_{i_1,...,i_p}^2 + \mu V(h) \right\} \tag{55}$$

and to treat this perturbatively in $\mu$.

$$
\begin{aligned}
S &= \left\{ \frac{\gamma}{N^{(p-1)/2}} \sum_{i_1,\dots,i_p} h_{i_1,\dots,i_p} \int d1\, j(1) \mathbf{s}_{i_1} \dots \mathbf{s}_{i_p}(1) + \frac{1}{2} \sum_{i_1,\dots,i_p} h^2_{i_1,\dots,i_p} + \mu V(h) \right\} \\
&= \left\{ \frac{\gamma^2}{N^{(p-1)}} \sum_{i_1,\dots,i_p} \mathbf{t}_{i_1,\dots,i_p} \mathbf{t}_{i_1,\dots,i_p} + \frac{1}{2} \sum_{i_1,\dots,i_p} (h_{i_1,\dots,i_p} + \mathbf{t}_{i_1,\dots,i_p})^2 + \mu V(h) \right\} \\
&= \left\{ \gamma^2 N \mathbf{I}^p + \frac{1}{2} \sum_{i_1,\dots,i_p} (h_{i_1,\dots,i_p} + \mathbf{t}_{i_1,\dots,i_p})^2 + \mu V(h) \right\} \\
&= \left\{ \gamma^2 N \mathbf{I}^p + \frac{1}{2} \sum_{i_1,\dots,i_p} (\tilde{h}_{i_1,\dots,i_p})^2 + \mu V(\tilde{h}_{i_1,\dots,i_p} - \mathbf{t}_{i_1,\dots,i_p}) \right\}.
\end{aligned}
\tag{56}
$$

Expanding the exponential of $V$, we obtain diagrams with contractions of $\tilde{h}^2_{i_1,\dots,i_p}$, and also lines with products of $\sum \mathbf{t}_{i_1,\dots,i_p}$, that may be expressed in terms of $Q(1,2)$'s. The same procedure, applied with replicas, yields the same diagrams, this time in terms of $Q_{ab}$.

As mentioned above, the value of corresponding dynamic and replica diagrams coincide $-\frac{d}{dj}\{\text{diagram}\}\big|_{j=0}$ give the same (a reparametrization-invariant fact) – if the dynamics has widely separated timescales, with one temperature $T(t, t') = T/X(C)$ per timescale, and $X(C) = x(q)$. Is the situation with timescales associated to different temperatures widely separated (a.k.a. time-ultrametricity) the only possibility for the coincidence of static and dynamics for diagrams? Our answer will be positive.

**Equality of temperatures and overlap**

In this section we shall review the argument for two sublattices $s_i$, $\sigma_i$ of a finite-dimensional spin system (not playing identical roles, i.e. there is no symmetry $\sigma \leftrightarrow s$), but it is valid for any two sets of variables such as spin and link energies. We consider the following correlations:

$$
q^{ss}_{ab} = \frac{1}{N} \sum_i \langle s_i \rangle_a \langle s_i \rangle_b \quad ; \quad q^{\sigma\sigma}_{ab} = \frac{1}{N} \sum_i \langle \sigma_i \rangle_a \langle \sigma_i \rangle_b \quad ; \quad q^{\sigma s}_{ab} = \frac{1}{N} \sum_i \langle \sigma_i \rangle_a \langle s_i \rangle_b \tag{57}
$$

and dynamically:

$$
\begin{aligned}
R^{ss}(t, t') &= \frac{1}{T^{ss}(t, t')} \frac{\partial}{\partial t'} C^{ss}(t, t') = \frac{X^{ss}(t, t')}{T} \frac{\partial}{\partial t'} C^{ss}(t, t'), \\
R^{\sigma\sigma}(t, t') &= \frac{1}{T^{\sigma\sigma}(t, t')} \frac{\partial}{\partial t'} C^{\sigma\sigma}(t, t') = \frac{X^{\sigma\sigma}(t, t')}{T} \frac{\partial}{\partial t'} C^{\sigma\sigma}(t, t'), \\
R^{\sigma s}(t, t') &= \frac{1}{T^{\sigma s}(t, t')} \frac{\partial}{\partial t'} C^{\sigma s}(t, t') = \frac{X^{\sigma s}(t, t')}{T} \frac{\partial}{\partial t'} C^{\sigma s}(t, t').
\end{aligned}
\tag{58}
$$

We now follow the same steps and apply the perturbations

$$
S = h_{i_1,\dots,i_p} j(1) [a s_{i_1} \dots s_{i_p} + b \sigma_{i_1} \dots \sigma_{i_p}] + h^2_{i_1,\dots,i_p}, \tag{59}
$$

for all $a, b$ and compute separately the corresponding generalized susceptibilities of each set of variables $I^{(p)}_{ss} = \int X^{ss}(C^{ss})[C^{ss}]^{(p-1)} dC^{ss}$, $I^{(p)}_{\sigma\sigma} = \int X^{\sigma\sigma}(C^{\sigma\sigma})[C^{\sigma\sigma}]^{(p-1)} dC^{\sigma\sigma}$ and $I^{(p)}_{\sigma s} = \int X^{\sigma s}(C^{\sigma s})[C^{\sigma s}]^{(p-1)} dC^{\sigma s}$. Equality of all makes us conclude that there is a correspondence between statics and dynamics at this partial level:

$$
x^{ss}(q^{ss}) \leftrightarrow X^{ss}(C^{ss}) \quad ; \quad x^{\sigma\sigma}(q^{\sigma\sigma}) \leftrightarrow X^{\sigma\sigma}(C^{\sigma\sigma}) \quad ; \quad x^{\sigma s}(q^{\sigma s}) \leftrightarrow X^{\sigma s}(C^{\sigma s}). \tag{60}
$$



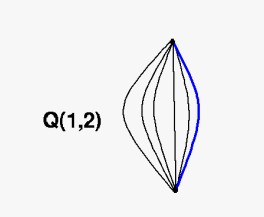

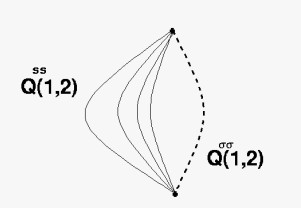

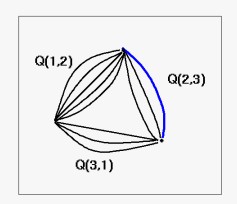

Figure 3: Three diagrams in terms of the superspace/replica order parameters. Each vertex stands for a superspace (or replica) index, each line for a $Q(a, b)$, where $(a, b)$ are the labels of the ends of the line.

**Thermalization and overlap equivalence**

We now show that if $C^{ss}(t, t') \to g[C^{\sigma\sigma}(t, t')]$ then $q^{ss} = g[q^{\sigma\sigma}]$ with the same function $g$, restricted to skeleton values of correlation associated with the Parisi ansatz. For non-skeleton values there can be no correspondence because the corresponding correlations, since they are absent from the static solution.

Construct now the susceptibilities via $h_{i_1,\dots,i_p;i_{p+1}} s_{i_1} \dots s_{i_p} (a\sigma_{i_{p+1}} + b s_{i_{p+1}})$ ($h_{i_1,\dots,i_p;i_{p+1}}$ is not symmetrized with respect to the last index) First, for $b = 0$ we get the middle diagram of figure 3:

$$I^{(p)} = \int dC^{\sigma\sigma} X^{\sigma\sigma}(C^{\sigma\sigma})[C^{ss}]^p + \int X^{ss}(C^{ss})C^{\sigma\sigma} d[C^{ss}]^p. \tag{61}$$

Now, $x^{ss}(q^{ss}), x^{\sigma\sigma}(q^{\sigma\sigma})$ are given by the statics too. *They are the same if there is overlap equivalence*. If so,

$$
\begin{aligned}
I^{(p)} &= \int \{dC^{\sigma\sigma}[C^{ss}]^p + d[C^{ss}]^p C^{\sigma\sigma}\} X^{ss}(C^{ss}) \\
&= X^{ss}(C^{ss})C^{\sigma\sigma}[C^{ss}]^p \big|_0^1 - \int dX^{ss} \{[C^{ss}]^p C^{\sigma\sigma}\} .
\end{aligned} \tag{62}
$$

This is equal to the equilibrium expression

$$
\begin{aligned}
I^{(p)} &= \int \{dq^{\sigma\sigma}[q^{ss}]^p + d[q^{ss}]^p q^{\sigma\sigma}\} x^{ss}(q^{ss}) = x^{ss}(q^{ss})q^{\sigma\sigma}[q^{ss}]^p \big|_0^1 - \int dx^{ss} \{[q^{ss}]^p q^{\sigma\sigma}\} \\
&= x^{ss}(q^{ss})q^{\sigma\sigma}[q^{ss}]^p \big|_0^1 - \int dq^{ss} \frac{dx^{ss}}{dq^{ss}} \{[q^{ss}]^p q^{\sigma\sigma}\} .
\end{aligned} \tag{63}
$$

The equality of all moments proves the equality of the functions $C^{\sigma\sigma} = g[C^{ss}]$ and $q^{\sigma\sigma} = g[q^{ss}]$ but only for the skeleton values at which $\frac{dx^{ss}}{dq^{ss}} \neq 0$.

Putting now $b \neq 0$ the linear term in $b$ implies:

$$
\begin{aligned}
I_b^{(p)} &= \int \{dC^{\sigma s}[C^{ss}]^p X^{\sigma s}(C^{\sigma s}) + d[C^{ss}]^p X^{ss}(C^{ss})C^{\sigma s}\} \\
&= X^{ss}(C^{ss})C^{\sigma s}[C^{ss}]^p \big|_0^1 - \int dX^{ss} \{[C^{ss}]^p C^{\sigma s}\} \\
&= x^{ss}(q^{ss})q^{\sigma s}[q^{ss}]^p \big|_0^1 - \int dq^{ss} \frac{dx^{ss}}{dq^{ss}} \{[q^{ss}]^p q^{\sigma s}\} .
\end{aligned} \tag{64}
$$

Again, the equality of all moments proves the equality of the functions $C^{\sigma s} = \hat{g}[C^{ss}]$ and $q^{\sigma s} = \hat{g}[q^{ss}]$ but only for the skeleton values at which $\frac{dx^{ss}}{dq^{ss}} \neq 0$.

**Three configurations**

This calculation was hinted at in FMPP, the only thing missing was a generating function for the diagram involved. We consider a slightly more complicated perturbation:

$$S = \frac{1}{2}\sum_{ij}\left(h_{ij}^{(1)}\right)^2 + \frac{1}{2}\sum_{ij}\left(h_{ij}^{(2)}\right)^2 + \frac{1}{2}\sum_{ij}\left(h_{ij}^{(3)}\right)^2$$
$$+ \frac{\gamma}{N}\sum_{ijk}\left(h_{ij}^{(1)}h_{kj}^{(2)} + h_{ij}^{(1)}h_{kj}^{(3)} + h_{ij}^{(2)}h_{kj}^{(3)}\right)s_k s_i, \tag{65}$$

leading to:

$$S = \frac{1}{2}\sum_{ij}\left(h_{ij}^{(1)}\right)^2 + \frac{1}{2}\sum_{ij}\left(h_{ij}^{(2)}\right)^2 + \frac{1}{2}\sum_{ij}\left(h_{ij}^{(3)}\right)^2$$
$$+ \frac{\gamma}{N}\sum_{ijk}\left(h_{ij}^{(1)}h_{kj}^{(2)} + h_{ij}^{(1)}h_{kj}^{(3)} + h_{kj}^{(2)}h_{ij}^{(3)}\right)\mathbf{t}_{ik}. \tag{66}$$

Integrating away the $h$'s, we get:

$$e^{\cdots + \gamma^2 N \int d1d2\, Q(1,2)j(1)j(2) + 2N\gamma^3 \int d1d2d3\, j(1)j(2)j(3)Q(1,2)Q(2,3)Q(3,1) + O(\gamma^4)}, \tag{67}$$

and similarly for replicas. More generally, denoting $I = i_1, \ldots, i_p$, $J = j_1, \ldots, j_r$, $K = k_1, \ldots, k_s$, $\mathbf{t}_I = \int j(1)s_{i_1}\ldots s_{i_p}\, d1$, and so on, we have, applying the corresponding perturbation:

$$S = \frac{1}{2}\sum_{IJ}\left(h_{IJ}^{(1)}\right)^2 + \frac{1}{2}\sum_{IJ}\left(h_{IJ}^{(2)}\right)^2 + \frac{1}{2}\sum_{JK}\left(h_{JK}^{(3)}\right)^2$$
$$+ \frac{\gamma}{N^{(3p-1)/2}}\sum_{IJK}\left(h_{IJ}^{(1)}h_{KJ}^{(2)} + h_{IJ}^{(1)}h_{KJ}^{(3)} + h_{KJ}^{(2)}h_{IJ}^{(3)}\right)\mathbf{t}_{KI}, \tag{68}$$

$$S =$$
$$\gamma^2 N\left(\int d1d2\, Q^{\bullet p}(1,2)j(1)j(2) + \int d1d2\, Q^{\bullet r}(1,2)j(1)j(2) + \int d1d2\, Q^{\bullet s}(1,2)j(1)j(2)\right)$$
$$+ N\gamma^3\int d1d2d3\, j(1)j(2)j(3)\left(Q^{\bullet p}(1,2)Q(2,3)^{\bullet s}Q^{\bullet r}(3,1) + Q^{\bullet p}(1,2)Q^{\bullet r}(2,3)Q^{\bullet s}(3,1)\right)$$
$$+ O(\gamma^4). \tag{69}$$

And similarly, for the replica calculation. This is the diagram to the right of Figure 3 Now, the Onsager property within a timescale implies that operators $Q^{\bullet r}$ and $Q^{\bullet s}$ commute, just as the Parisi matrices do. The result of the diagram is reported in FMPP and is proportional to:

$$\int dq_{12}dq_{23}dq_{31}\, P(\text{triangle})\, (q_{12})^p(q_{23})^r(q_{31})^s. \tag{70}$$

Equality of these for all $p, r, s$ implies the equality of the probability of triangles constituted by skeleton values of overlaps (we note again that the Parisi scheme has nothing to say about triangles formed by intermediate values of correlations 'within a scale').

## 5 The role of reparametrization invariance(s)

As mentioned in the introduction, connecting two independent systems with a small local interaction $\sum_i s_i\sigma_i$, so that the $\sigma_i, s_i$ become sublattices of a single system, puts the system (and us)

in a dilemma: if the systems had different effective temperatures in a same timescale, the combined system would violate the scenario we have been describing, including the Parisi scheme, because it will have different temperatures for different observables in the same timescale. Thus, the new coupled system will immediately fall outside the scheme. Something clearly is amiss, because this would happen even for mean-field models, for which we know that the scenario holds.

One possibility is that all imaginable systems that satisfy a multithermalzed/RSB scheme have the same timescales at the same values of $T$ for the same $X$. Thus, any two systems evolving at the same temperatures would have automatically all the effective temperatures at the same scales. There is a well-understood counterexample to this possibility: ferromagnetic domain growth has fast timescale -relaxation within domains – and a a slow timescale, correponding to the displacements of domain walls. The effective temperature for the slow motion is infinite [34]. Now, the slow timescale is not universal: it may be $C = C\left(\frac{t'}{t}\right)$ for a pure ferromagnet, or $C = C\left(\frac{\ln t'}{\ln t}\right)$ for a 'dirty' one.

The other possibility, already hinted at years ago [10,31,32], is as follows: when the interaction is strong enough, the two systems change their temperatures so they become equal. This requires a certain critical coupling strength. When the coupling is weaker than that, something stranger should happen: the timescales associated with the two temperatures "push each other apart", so the combined system has one more timescale, and the scenario is recovered. That this should happen even with very weak interaction is only possible because the systems develop independent reparametrization invariances, and coupling is always relevant. This very surprising phenomenon 'saves the scenario'.

In Ref. [11, 12] we have studied in detail a slightly different context in which this may happen: instead of coupling a system to another one, we couple it to a 'multibath'.

## 5.1  How do the families of reparametrization invariances come about

In mean-field models – in fact the only systems for which we *know* that the scenario holds, the reparametrization invariant families come about as follows. One arrives with the usual formalism for dynamic equations for correlations and response function, either by summing ladder diagrams or by working taking saddle point in the dynamic path integral averaged over disorder.

One then verifies that each scale may be treated separately, with the faster scales acting as if they were instantaneous and the slower ones as being frozen. The first step, shared by SYK (which has only two scales) involves separating out the fastest scale, the only one in which time-derivatives are relevant. Then, one considers the infrared scale which will have a reparametrization invariance in the slow-dynamics limit in which the time-derivatives may be neglected.

In some systems the procedure stops here. However, in systems such as the Sherrington-Kirkpatrick model, there is a 'more infrared' scale, which is infinitely slower than the previous one, and may be separated likewise, and then another, and another. Each scale may be time-reparametrized freely provided it remains separated from the previous one.

## 5.2  Two glasses and a wormhole

Coupling systems with reparametrization invariances is generally interesting, because the coupling will almost surely be relevant, since relative reparametrizations between systems are soft. In a series of papers [35, 36], two SYK models – toy versions of Black Holes – have been coupled, and the effect is a system with a combined first order transition line with hysteresis in the temperature-coupling plane, terminating in a triple point. Let us briefly show how very much

the same transition is expected to happen when coupling two glasses, for the same reasons. A direct interpretation, not using reparametrization invariance, is available in the glassy case.

Let us recall a connection between stochastic and quantum dynamics that has been already used several times in the past in statistical physics, condensed matter and quantum field theory [37–39] and which we have exploited to lay a bridge between glasses and quantum systems with large $T = 0$ entropies [40]. Just as above we consider two systems $N$ coupled degrees of freedom $s_i(t), \sigma_i(t)$ evolving by stochastic Langevin dynamics

$$
\begin{aligned}
\dot{s}_i(t) &= -\frac{\partial V}{\partial s_i} + \eta_i(t), \\
\dot{\sigma}_i(t) &= -\frac{\partial V}{\partial \sigma_i} + \tilde{\eta}_i(t),
\end{aligned}
\tag{71}
$$

and $V$ is the interaction potential, which we shall take to be:

$$
V = \sum_{ijk} J_{ijk} \left( s_i s_j s_k + \sigma_i \sigma_j \sigma_k \right) + z \left( \sum \sigma_i^2 - N \right) + \tilde{z} \left( \sum s_i^2 - N \right),
\tag{72}
$$

where the $J$ are random and fully-connected and the terms proportional to $z$ impose a spherical constraint $\sum_i \sigma_i^2 = \sum_i s_i^2 = N$. This is the simplest and better understood mean-field glass, but there are plenty of other examples in the literature, with and without disorder. Here $T_s$ is the (classical) temperature of the thermal bath to which the system is coupled, and $\eta_i(t), \tilde{\eta}(t)$ are a Gaussian white noises with covariance $\langle \eta_i(t)\eta_i(t') \rangle = 2T_s \delta(t - t')$.

The evolution of the probability density is generated by the Fokker–Planck operator $H_{\text{FP}}$,

$$
\partial_t P_t(\mathbf{q}) = \sum_i \frac{\partial}{\partial q_i} \left[ T_s \frac{\partial}{\partial q_i} + \frac{\partial V}{\partial q_i} \right] P_t(\mathbf{q}) \equiv -H_{\text{FP}} P_t(\mathbf{q}),
\tag{73}
$$

where $q_i = \{s, \sigma\}$. Detailed balance allows us to write this in an explicitly Hermitian form [39, 41]. Rescaling time, one can define the operator

$$
H = \frac{T_s}{2} e^{V/2T_s} H_{\text{FP}} e^{-V/2T_s} = \sum_i \left[ -\frac{T_s^2}{2} \frac{\partial^2}{\partial q_i^2} + \frac{1}{8} \left( \frac{\partial V}{\partial q_i} \right)^2 - \frac{T_s}{4} \frac{\partial^2 V}{\partial q_i^2} \right].
\tag{74}
$$

$H$ has the form of a Schrodinger operator with $T_s$ playing the role of $\hbar$, unit mass and potential

$$
V_{\text{eff}} = \frac{1}{8} \left( \frac{\partial V}{\partial q_i} \right)^2 - \frac{T_s}{4} \frac{\partial^2 V}{\partial q_i^2}.
\tag{75}
$$

The spectrum of eigenvalues $\lambda_i$ and eigenvectors $\psi_i$ of $H$ (or $H_{\text{FP}}$) have a direct relation to metastable states of the original diffusive dynamics ([42, 43], see also [44]):

- The equilibrium state has $\lambda_o = 0$ and the corresponding right eigenvector of $H_{\text{FP}}$ is the Boltzmann distribution associated with the energy function $V$.

- Given a timescale $t^*$, the number of eigenvectors with $\lambda_i < \frac{1}{t^*}$ is the number of metastable states of the diffusive model with lifetime larger than $t^*$. *In particular, the eigenvalues $\lambda_i \to 0$ in the thermodynamic limit correspond to metastable states whose lifetime diverges with $N$.*

- Hence, the resulting object $\mathcal{N}(\beta_q) = \text{Tr}\, e^{-\beta_q H_{\text{FP}}} = Z(\beta_q)$ counts the number of states of the system that are stable up to a time $\beta_q$ or longer [44].

We thus have introduced a "quantum" Hamiltonian $H$, which is associated with a quantum temperature $T_q = 1/\beta_q$. The original temperature $T_s$ now plays the role of the quantum parameter, $\hbar$. Our "quantum energy" is associated with the eigenvalues of $H$, which are a measure of the lifetimes of the original classical diffusive system. We may analize this 'quantum system' in terms of the underlying glassy model. The extensive 'zero temperature entropy' is nothing but the log of the number of metastable states, the 'glassy' reparametrization invariance is now quite analogous to the one of SYK [40] We now couple the two systems through a term:

$$H_\mu = H - \mu \sum_i \sigma_i s_i, \tag{76}$$

which no longer corresponds to a purely stochastic evolution, but rather to the dynamic large deviation of $\int dt \sum_i \sigma_i s_i$, a generator function. The system will thus have larger than zero eigenvalue ground state, the value being precisely the large-deviation function for each $\mu$. Had we coupled the system at the level of $V$, the joint system would have zero energy quantum ground state: we know this because the system so obtained is still a glass.

The partition function of the Hamiltonian $H_\mu$ reads

$$Z(\mu, \beta_q) = \operatorname{Tr} e^{-\beta_q H_\mu} = \operatorname{Tr} e^{-\frac{1}{2}\beta_q [T_s H_{\mathrm{FP}} - \mu \sum_i \sigma_i s_i]} = \int dq\, e^{-\mu N Q} \mathcal{N}(Q, \beta_q), \tag{77}$$

where $\mathcal{N}(Q, \beta_q)$ measures the number of pairs of metastable states at mutual distance $N\beta_q Q = \int_0^{\beta_q} dt'\, s_i(t')\sigma_i(t')$. For the two coupled systems two phenomena compete: there is an exponential number of metastable states in each system, all of them (for large $N$) marginal in the sense of having gapless vibration spectra. The metastable states of the combined system is the set of pairs of states one in each system, and is overwhelmingly dominated by taking different states in each subsystem – these pairs will almost all have small mutual overlap, for entropic reasons. An attractive interaction between configurations of subsystems privileges on the contrary choosing the same state in both subsystems. Bearing in mind that an energetic term dominates the entropic term at lower temperatures, we get a first order mechanism, see Figure 4.

Let us conclude this section with a remark. When we construct a 'quantum' Hamiltonian à la Rokhsar-Kivelson, the usual imaginary-time partition function corresponds, as we have mentioned, to counting the number of metastable states; from the point of view of the stochastic system, counting periodic stochastic trajectories. The *real time* evolution (with an '*i*') does not have any clear meaning from the stochastic point of view. Finally, the 'aging' solution corresponds to the following construction: for a general Hamiltonian $H$, given its ground-state $|0\rangle$ with eigenvalue $\lambda_0$, and a random initial state $|\text{init}\rangle$, one computes correlations with the propagation

$$\langle A \rangle_{aging}(t) = e^{\lambda_0 t} \langle 0|Ae^{-tH}|\text{init}\rangle/\langle 0|\text{init}\rangle; \quad C_{aging}(t, t') = e^{\lambda_0 t} \langle 0|Ae^{-(t-t')H}Ae^{-t'H}|\text{init}\rangle/\langle 0|\text{init}\rangle.$$

In a quantum system with ground-state entropy this process only becomes stationary in times $t'$ that diverge with $N$.

# 6 Conclusion

In this paper we discuss the essential role of time-reparametrization quasi-invariances in solutions of glassy dynamics and equilibrium. As an intermediate step, we have needed to complete the program of Franz et al (FMPP) in establishing a direct link between Parisi scheme and dynamic 'multithermalization', valid for finite-dimensional systems under the assumption

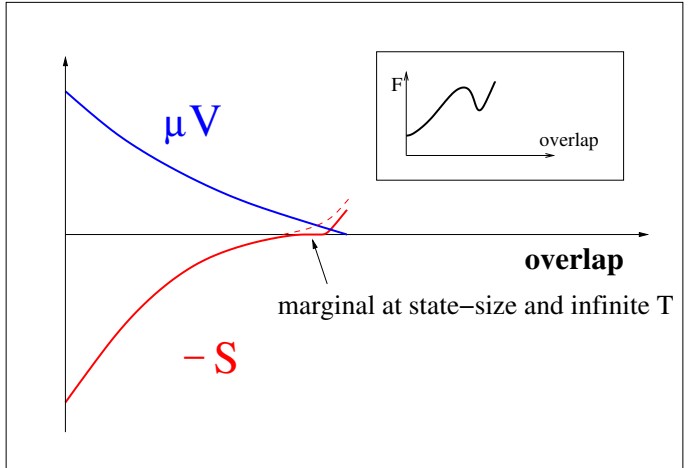

Figure 4: The Franz-Parisi potential [45, 46]: complexity (the log number of metastable states at a given overlap) and interaction energy, both plotted in terms of the overlap. At the point in which the overlap coincides with the state size, the entropy is the one of a single system, and the point is marginal. At larger overlaps, i.e within a state, the complexity becomes negative. The dashed line represents the correction coming from counting states with finite lifetime, as one must at finite "quantum temperature". The sum of the attraction and entropic effects yields a first-order transition mechanism (inset).

of stochastic stability with respect to random, long range interactions. For this we had to show that static and dynamic ultrametricies imply one another. In view of the results in mathematical physics [47, 48] systems that are stochastically stable with respect to long-range random correlations should have static and dynamic properties corresponding to the scenario discussed here. If a system still has a glassy phase, but does not correspond to this scenario, then it seems one would have to conclude that symmetries are more broken (smaller residual groups), in an at present unknown way.

An intriguing possibility concerns the quantum SYK-like systems. These have a single infrared timescale, which diverges as the inverse temperature. The analogy with spin-glasses suggests that variants with nested divergent timescales should also be possible.

## Acknowledgements

I wish to thank F. Corberi and S Franz for clarifying conversations, and especially F Camilli, PL Contucci and E Mingione for pointing out an error in the first version of the manuscript. This work is supported by the Simons Foundation Grant No 454943.

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
