# Peer review of "Time-reparametrization invariances, multithermalization and the Parisi scheme"

_SciPost Physics, doi:SciPost Phys. Core 6, 001 (2023)_

## Round 3 · Referee Report · Anonymous (Referee 1) · 2021-6-26

Strengths

1) The paper gives a clear argument that the Parisi equilibrium solution and the multithermalization scenario for the dynamics in glassy systems and those with slow dynamics are not only closely related, but in fact imply each other, and that reparametrization symmetry is the organizing principle.

2) The arguments in the paper rely on a number of technical details, but the paper is written so as to keep the main ideas at the forefront at all times.

Weaknesses

1) The reliance on arguments made in Ref. [14] mean that the work is less accessible to a reader who is not familiar with that work than might otherwise be the case.

Report

It has been known for quite some time that there is a close connection between the multithermalization dynamics in models of glassy systems and the corresponding equilibrium solutions to the same models. An important aspect of the dynamics is that there are widely separated timescales in the dynamics and within a timescale, observables share the same temperature. Observables at different timescales can have different temperatures. In this manuscript, Kurchan argues that these two scenarios imply each other - the consistency arises because of a reparametrization symmetry. This work tightens the links between equilibrium and dynamics in glassy systems, using reparametrization symmetry as the organizing principle to do so.

In my opinion this is nice work and meets the standards for publication in SciPost Physics, but I think there is room for improvement in the presentation and there are some points I think the author should check (many of which I think are typos), before this work is ready for publication.

Requested changes

1) In the last bullet point under Sec. II.A, would it be more accurate to write $h(t) = \ln t$ rather than $h(t) \to \ln t$?

2) I think there may be a typo in the inequality on the right hand side of Eq. (18).

3) In the second line of Eq. (38), should the first term be $h_{ij}^{(1)}h_{jk}^{(2)} s_i^{(1)} s_k^{(2)}$ as opposed to $h_{ij}^{(1)}h_{jk}^{(2)} s_k^{(1)} s_i^{(2)}$? This would seem to be more consistent with the other two terms on that line.

4) In Eq. (39), it would be helpful to include the summation indices (presumably $ijk$). In the expression for $C^{(4)}$ should there be the correlation $\left<s_i^{(1)}(t) s_k^{(4)}(t^\prime)\right>_\gamma$? Also, in the expression for $R^{(4)}$, normally the response is defined in terms of the derivative of a $s_i$ with respect to a field rather than another $s_k$. This seems to be an unusual definition for a response, or is it a typo?

5) Between Eqs. (45) and (46) the assertion is made that $\hat{C}^{(n)}(\omega)$ and $\hat{\chi}_s^{(n)}(\omega)$ are peaked around zero - presumably $\omega = 0$. Please give a little more explanation for why this is the case.

6) When I compare Eq. (48) to the similar expression in Ref. [7], the corresponding susceptibility in Ref. [7] is defined through a derivative with respect to a field, rather than $\gamma$. Could you explain why this is.

7) Under Eq. (52), I don't understand the notation for $j(a)$, since the definition of $j(a)$ includes an index $b$, yet there is no summation over index $b$ in Eq. (52). Perhaps this notation needs to be tidied up a bit?

8) In Eq. (56) $\mu$ appears to have been dropped in the second and third lines of the equation.

9) At the bottom of page 12, there appears to be something missing in the sentence "For non-skeleton values there can be no correspondence because the corresponding correlations, since they are absent from the static solution."

10) I found Figure 3 somewhat confusing. It would be helpful to give a more detailed caption.

11) I was confused in the step between Eq. (65) and Eq. (66). Is there an omitted summation over superspace variables in Eq. (65)?

12) In Eq. (69) is there a distinction between $Q^{\bullet p}(1,2)$ and $Q(1,2)^{\bullet p}$? If not, then it would be helpful to write all the $Q$s with the same notation.

13) In Eq. (70), should it be $(dq_{23})^r (dq_{31})^s$ or $(q_{23})^r (q_{31})^s$? The latter would seem to be more compatible with Ref. [14].

  • validity: high
  • significance: high
  • originality: high
  • clarity: good
  • formatting: good
  • grammar: good

Author:  Jorge Kurchan  on 2022-09-06  [id 2790]

(in reply to Report 1 on 2021-06-26)

I wish to thank the referee for a thorough reading. I have corrected the mistakes, and added some phrases where necessary.

1) done 2) done 3) done 4) done

5) Between Eqs. (45) and (46) the assertion is made that ^C(n)(ω) and ^χ(n)s(ω) are peaked around zero - presumably ω=0. Please give a little more explanation for why this is the case.

I have added the following phrase:

"We have thus constructed a whole set of pairs a set $R^{(n)}$, $C^{(n)}$ starting from $C$, $R$: is it possible that they are related by the same $T_{eff}$$ at equal times? It turns out that it is much easier to answer this question by comparing $n=1$ with large $n$, this result on its own will tell us that a necessary condition is to have timescale separation."

and after (45) we modified the phrase to a clearer form:

If we consider a large value of $n$, then $\hat C^{(n)}(\omega)$ and ${\hat \chi_s^{(n)}(\omega)}$ will be peaked around some value of $\omega$ dominated by the maximum of $R(\omega)$, which for a strongly dissipative system we expect to be zero. Then we may write:

6) When I compare Eq. (48) to the similar expression in Ref. [7], the corresponding susceptibility in Ref. [7] is defined through a derivative with respect to a field, rather than γ. Could you explain why this is.

The expressions are related to leading order, because only terms with paired $h_{i_1,...,i_p}$ survive.

7) Under Eq. (52), I don't understand the notation for j(a), since the definition of j(a) includes an index b, yet there is no summation over index b in Eq. (52). Perhaps this notation needs to be tidied up a bit?

this is the analogue of the standard practice to compute the energy at time t: one adds j= 1+ j E(t) *\delta (t-t_0) as a source. When times are doubled, in fact only the linear term in j survives, and the other time is integrated without a source.

8) corrected (thanks)

9) At the bottom of page 12, there appears to be something missing in the sentence "For non-skeleton values there can be no correspondence because the corresponding correlations, since they are absent from the static solution."

10) I found Figure 3 somewhat confusing. It would be helpful to give a more detailed caption.

I addeed a clarification in the caption.

11) I was confused in the step between Eq. (65) and Eq. (66). Is there an omitted summation over superspace variables in Eq. (65)?

the $t_{i...}$ are themselves already summed over superspace/replicas

12) corrected

13) corrected

---

## Round 3 · Referee Report · Anonymous (Referee 2) · 2021-6-30

Strengths

The intriguing connection between the effective temperature function $X(C)$ discovered by the dynamical mean-field theory for glassy systems and the Parisi's order parameter function $x(q)$ obtained earlier in the static mean-field theory has attracted researchers for a long time. The two functions govern physical quantities in the dynamics and statics. The present paper is remarkable because of the following.

  1. A theoretical argument is constructed which implies separations of timescales is not only sufficient but also necessarily for the agreement of the two functions in finite dimensions.

  2. An interesting physical consequence is predicted: time-reorganization.

Weaknesses

There are some points that need to be considered to improve the manuscript as stated below.

Report

The paper is very interesting and suggestive so that it should be published. However the following points need to be considered for improvements.

Requested changes

[A]. For 'thermalization', the author considers two rather different settings: (i) thermalization of 'spins' (microscopic degrees of freedom) at different parts of a single (big, statistically homogeneous) system (ii) thermalization of two glasses (which can be very different) brought in contact. In [13,14] FMPP (Franz-Mezard-Parisi-Peliti) considered only the setting (i). The author uses these two settings in a mixed way.

The setting (ii) is very interesting by itself but I think the paper becomes more logical if these are used/discussed separately. For instance, section IV can be formulated coherently using exclusively setting (I). Various possibilities in the setting (ii) may be discussed in a new section dedicated to it.

For (i), one expects the environments of the spins at different parts of the same 'homogeneous' system to be just the same in a statistical sense. Thus it is natural to assume the same Parisi's ansatz for them. On the other hand, for (ii), it is not obvious what one should expect in general. For instance, the correct choice of Parisi's ansatz for the two glass problems discussed in sec V B is not obvious. The combined system may behave as one thermalized body or not depending, for instance on the strength of mutual coupling.

In sec IV A, setting (i) is used. It is shown that separation of times, which implies wide enough time scales such that $X(t,t;)$ remain essentially constant in each time scale, is essential to establish a common effective temperature among different parts of the same system in a time scale. The argument based on 'convolution' (Fig 2) is convincing. However, it is not obvious how to rephrase this argument in (ii).

In sec IV B, after (57), the setting (ii) is used. To go from (61) to (62), it is assumed that (a) $X^{\sigma\sigma}C^{\sigma\sigma})=X^{ss}(C^{ss})$ (thermalization of two glass in a time scale) holds. Then in (63) it is assumed that (b) $x^{\sigma\sigma}(q^{\sigma\sigma})=x^{ss}(q^{ss})$ (same Parisi's ansatz for two glass in a correlation scale) holds. These assumptions may or may not hold. However, they would be justified if one rephrases the same argument in setting (i) because the argument in sec IV A for the dynamics applies and it is natural to expect for the statics that the same Parisi's ansatz applies for different parts of the same 'homogeneous' system. For the setting (ii), the argument implies that if the assumption (a) fails, (b) fails also. This is an interesting point but maybe discussed separately.

[B] What is exactly the behavior of the two glasses discussed in sec V B in the case of weak coupling? For strong enough coupling the two glasses thermalize with a common effective temperature [31]. The time-reorganization phenomenon takes place here when the coupling is weak? Or the simple 1RSB system cannot make it?

Minor points (1) p4. 1st paragraph. Fig. II A $\to$ Fig. 1. (2) p6. (17) if $C(t_{\rm max},t_{\rm int}) \leq$ '' $\to$if $C(t_{\rm min},t_{\rm int}) \leq$''. (3) p6. (18) if $C(t_{\rm max},t_{\rm int})\leq$ '' $\to$if $C(t_{\rm min},t_{\rm int}) \leq$''. (4) p6. sec D. The 1st sentence. Th properties $\to$ The properties''. (5) p8, (38). In the last term, may be $h_{ij}^{(4)}h_{jk}^{(1)}s^{(4)}_{i}s^{(1)}_{k}$ is missing? (6) p9, (40) and (41) $\gamma^{3} \to \gamma^{4}$?. (7) p9. I think what is written in the caption of Fig. 2 should be brought to the main text and placed at the end of sec IV A because it is a crucial point in the present paper. If I understood correctly,separation of timescales'' implies wide enough time scales'' such thatthe effective temperature remains essentially constant within such a time scale.'' And this is crucial to ensure thermalization among different components of a system within such a time scale. (8) p13, below (61), Now $x(q^{ss}),x(q^{\sigma\sigma})$'' $\to$ $x^{ss}(q^{ss}),x^{\rm \sigma\sigma}(q^{\sigma\sigma})$''. - I think this sentence should be placed below (62). - I do not know what one can say aboutoverlap equivalence'' of two different glasses. Actually, I do not understand what the term ``overlap equivalence'' means in this context. - As stated above I believe the use of setting (i) is much better here. (9) p16, see Fig. VB $\to$ Fig. 4.

  • validity: high
  • significance: high
  • originality: high
  • clarity: good
  • formatting: good
  • grammar: good

Author:  Jorge Kurchan  on 2022-09-06  [id 2791]

(in reply to Report 2 on 2021-06-30)

I wish to thank the referee for a very insightful mreading. I have followed all their minor points (see below). The major remark, is however, the following:

" [A]. For 'thermalization', the author considers two rather different settings: (i) thermalization of 'spins' (microscopic degrees of freedom) at different parts of a single (big, statistically homogeneous) system (ii) thermalization of two glasses (which can be very different) brought in contact. In [13,14] FMPP (Franz-Mezard-Parisi-Peliti) considered only the setting (i). The author uses these two settings in a mixed way.

The setting (ii) is very interesting by itself but I think the paper becomes more logical if these are used/discussed separately. For instance, section IV can be formulated coherently using exclusively setting (I). Various possibilities in the setting (ii) may be discussed in a new section dedicated to it.

For (i), one expects the environments of the spins at different parts of the same 'homogeneous' system to be just the same in a statistical sense. Thus it is natural to assume the same Parisi's ansatz for them. On the other hand, for (ii), it is not obvious what one should expect in general. For instance, the correct choice of Parisi's ansatz for the two glass problems discussed in sec V B is not obvious. The combined system may behave as one thermalized body or not depending, for instance on the strength of mutual coupling.

In sec IV A, setting (i) is used. It is shown that separation of times, which implies wide enough time scales such that X(t,t;) remain essentially constant in each time scale, is essential to establish a common effective temperature among different parts of the same system in a time scale. The argument based on 'convolution' (Fig 2) is convincing. However, it is not obvious how to rephrase this argument in (ii).

In sec IV B, after (57), the setting (ii) is used. To go from (61) to (62), it is assumed that (a) XσσCσσ)=Xss(Css) (thermalization of two glass in a time scale) holds. Then in (63) it is assumed that (b) xσσ(qσσ)=xss(qss) (same Parisi's ansatz for two glass in a correlation scale) holds. These assumptions may or may not hold. However, they would be justified if one rephrases the same argument in setting (i) because the argument in sec IV A for the dynamics applies and it is natural to expect for the statics that the same Parisi's ansatz applies for different parts of the same 'homogeneous' system. For the setting (ii), the argument implies that if the assumption (a) fails, (b) fails also. This is an interesting point but maybe discussed separately.

[B] What is exactly the behavior of the two glasses discussed in sec V B in the case of weak coupling? For strong enough coupling the two glasses thermalize with a common effective temperature [31]. The time-reorganization phenomenon takes place here when the coupling is weak? Or the simple 1RSB system cannot make it? "

ANSWER: The referee has a point. I have modified section IV (b) to the same approach of a single system, and have tried to make it that way in all but the last (wormhole) section.

Minor points (1) p4 done (2) p6 done (3) p6.done (4) p6. done (5) p8, (38). In the last term, may be h(4)ijh(1)jks(4)is(1)k is missing? answer: see below (6) p9, (40) and (41) γ3→γ4? answer: I chose to compute an expectation rather than a derivative , so one of the γ is not there (7) clarified (8) done (9) done

---

## Round 4 · List of Changes

All typos corrected. Clarifications requested added.

---

## Editorial Decision

published